# Embryonic Stage of Congenital Zika Virus Infection Determines Fetal and Postnatal Outcomes in Mice

**DOI:** 10.3390/v13091807

**Published:** 2021-09-11

**Authors:** Eri Nakayama, Yasuhiro Kawai, Satoshi Taniguchi, Jessamine E. Hazlewood, Ken-ichi Shibasaki, Kenta Takahashi, Yuko Sato, Bing Tang, Kexin Yan, Naoko Katsuta, Shigeru Tajima, Chang Kweng Lim, Tadaki Suzuki, Andreas Suhrbier, Masayuki Saijo

**Affiliations:** 1Department of Virology I, National Institute of Infectious Diseases, Tokyo 162-8640, Japan; rei-tani@nih.go.jp (S.T.); shi-k@nih.go.jp (K.-i.S.); nkatsuta@nih.go.jp (N.K.); stajima@nih.go.jp (S.T.); ck@nih.go.jp (C.K.L.); msaijo@nih.go.jp (M.S.); 2Management Department of Biosafety and Laboratory Animal, Division of Biosafety Control and Research, National Institute of Infectious Diseases, Tokyo 162-8640, Japan; kaya@nih.go.jp; 3Inflammation Biology Group, QIMR Berghofer Medical Research Institute, Brisbane, QLD 4029, Australia; Jessamine.Hazlewood@qimrberghofer.edu.au (J.E.H.); Bing.Tang@qimrberghofer.edu.au (B.T.); Kexin.Yan@qimrberghofer.edu.au (K.Y.); Andreas.Suhrbier@qimrberghofer.edu.au (A.S.); 4Department of Pathology, National Institute of Infectious Diseases, Tokyo 162-8640, Japan; tkenta@niid.go.jp (K.T.); kiyonaga@nih.go.jp (Y.S.); tksuzuki@niid.go.jp (T.S.)

**Keywords:** Zika virus, congenital Zika syndrome, SHIRPA, microcephaly, sequelae, trimester, mouse model

## Abstract

Zika virus (ZIKV) infection during pregnancy causes a wide spectrum of congenital abnormalities and postnatal developmental sequelae such as fetal loss, intrauterine growth restriction (IUGR), microcephaly, or motor and neurodevelopmental disorders. Here, we investigated whether a mouse pregnancy model recapitulated a wide range of symptoms after congenital ZIKV infection, and whether the embryonic age of congenital infection changed the fetal or postnatal outcomes. Infection with ZIKV strain PRVABC59 from embryonic day 6.5 (E6.5) to E8.5, corresponding to the mid-first trimester in humans, caused fetal death, fetal resorption, or severe IUGR, whereas infection from E9.5 to E14.5, corresponding to the late-first to second trimester in humans, caused stillbirth, neonatal death, microcephaly, and postnatal growth deficiency. Furthermore, 4-week-old offspring born to dams infected at E12.5 showed abnormalities in neuropsychiatric state, motor behavior, autonomic function, or reflex and sensory function. Thus, our model recapitulated the multiple symptoms seen in human cases, and the embryonic age of congenital infection was one of the determinant factors of offspring outcomes in mice. Furthermore, maternal neutralizing antibodies protected the offspring from neonatal death after congenital infection at E9.5, suggesting that neonatal death in our model could serve as criteria for screening of vaccine candidates.

## 1. Introduction

The World Health Organization declared the outbreak of Zika virus (ZIKV) infection in the American continent as a public health emergency of international concern in 2016. ZIKV causes a spectrum of congenital abnormalities including fetal loss, intrauterine growth restriction (IUGR), neonatal death, and microcephaly, together termed congenital Zika syndrome (CZS), which is likely associated with complex and life-long disabilities in children born to women infected with ZIKV during pregnancy [1,2,3,4,5,6]. Postnatal developmental sequelae, such as gross motor impairment, delayed neurodevelopment, cognitive impairment, auditory abnormalities, and/or ophthalmological abnormalities have also been recognized after congenital ZIKV infection, [7,8,9,10,11,12,13,14,15,16]. Some children who were asymptomatic with normal head circumferences at birth also developed postnatal symptoms [7,8,14,17,18]. CZS or developmental sequelae was recognized regardless of the trimesters in which the pregnant mothers were infected [1,7,18,19,20], although ZIKV infection in the first trimester has been thought to be a risk factor for severe CZS [21,22,23,24,25]. In line with this, a series of murine pregnancy models have been established [26,27,28,29,30,31,32,33,34,35,36,37]; however, each model recapitulated only some of the fetal or postnatal symptoms seen in human cases (Appendix A). Moreover, the embryonic ages of congenital ZIKV infection and inspection of offspring, virus strain, infection dose, or route of infection were not consistent in the mouse models (Appendix A), and how these differences affect the outcomes remains elusive. In this study, IFNα/β receptor knockout (IFNAR^−/−^) dams that were crossed with wild-type sires were infected subcutaneously (s.c.) with ZIKV at various embryonic days, and a series of fetal and postnatal outcomes were comprehensively evaluated.

## 2. Materials and Methods

### 2.1. Ethics Statement

All mouse experiments were conducted in accordance with the Guidelines for Animal Experiments performed at the National Institute of Infectious Diseases (NIID) or the Australian Code for Care and Use of Animals for Scientific Purposes, as outlined by the National Health and Medical Research Council of Australia. Animal experiments were approved by the Animal Welfare and Animal Care Committee of NIID (Ethics numbers: 116123 and 119155) or the QIMR Berghofer Medical Research Institute Animal Ethics Committee (Ethics number: A1604-611M). All mice were bred and housed under specific pathogen-free conditions.

### 2.2. Cell and Virus Stocks

Vero (strain 9013, JCRB9013, the Japanese Collection of Research Bioresources Cell Bank, Osaka, Japan) and C6/36 cells (CRL1660, the American Type Culture Collection Manassas, VA, USA) were maintained in Eagle’s minimum essential medium (MEM) supplemented with 10% fetal bovine serum (FBS) and 100 μg/mL penicillin-streptomycin (Life Technologies, Carlsbad, CA, USA). Vero cells and C6/36 cells were cultured at 37 °C and 28 °C, respectively, in a 5% CO_2_ atmosphere. ZIKV strain PRVABC59 (GenBank accession no. KU501215), which was isolated from a patient in Puerto Rico in 2015 [38], was kindly provided by Dr. Beth Bell of the US Center for Disease Control and Prevention. E protein amino acid position 330 of PRVABC59 was a mixture of V and L, as previously reported [39,40]. Natal RGN strain (GenBank accession no. KU527068) was isolated from human fetal autopsy cases with microcephaly in Brazil and prepared as previously described [26,41,42]. ZIKV stocks were tittered by plaque assay on Vero cells, as described previously [43,44].

### 2.3. Virus Titration

Indicated tissues and serum obtained from the blood of the tail vein were collected at the specified time points and stored at −80 °C until analysis. The tissues were homogenized in MEM containing 2% FBS (2MEM) using a tissue homogenizer and beads (Bio Medical Science, Tokyo, Japan) according to the manufacturer’s instructions. The 50% cell culture infective dose (CCID_50_) assays for serum and supernatants from homogenized tissues were performed as described previously [26,40,41,42,43,45,46].

### 2.4. Reverse Transcription Quantitative PCR (qRT-PCR)

qRT-PCR was performed as described previously [40,42,45,46]. Briefly, tissues were placed in RNAlater (Ambion, Austin, TX, USA), and RNA was extracted with TRIzol (Life Technologies, Carlsbad, CA, USA) from homogenized tissues prepared by bead homogenization. cDNA was generated using an iScript cDNA Synthesis Kit (Bio-Rad, Hercules, CA, USA). qPCR was performed using iTaq Universal SYBR Green Supermix (Bio-Rad) and the following primers: ZIKV E-Forward, 5′-CCGCTGCCCAACACAAG-3′; ZIKV E-Reverse, 5′-CCACTAACGTTCTTTTGCAGACAT-3′; ZIKV prM-Forward, TTGGTCATGATACTGCTGATTGC-3′; and ZIKV prM-Reverse, 5′- CCTTCCACAAAGTCCCTATTGC-3′ [47]. Values were normalized using the housekeeping gene mouse RPL13A (Forward, 5′-GAGGTCGGGTGGAAGTACCA-3′; Reverse, 5′-TGCATCTTGGCCTTTTCCTT-3′) [48,49].

### 2.5. Histology and Immunohistochemistry (IHC)

Tissue samples were fixed in 10% phosphate-buffered formalin, embedded in paraffin, sectioned, and stained with hematoxylin and eosin (H&E). IHC was performed using an anti-ZIKV NS1 antibody (C01886G, Meridian Bioscience, Cincinnati, OH, USA) as the primary antibody [40,43]. Specific antigen-antibody reactions were visualized by 3,3-diaminobenzidine tetrahydrochloride staining using a VECTASTAIN ABC HRP system (Vector Laboratories, Burlingame, CA, USA).

### 2.6. Mice

IFNAR^−/−^ mice on a C57BL/6J background were bred in-house at NIID [40,43]. Female IFNAR^−/−^ mice (>7 weeks old) were paired with C57BL/6J mice (>8 weeks old) purchased from SLC Ltd. (Shizuoka, Japan), as described previously [29]. When a plug was detected, this was deemed embryonic day 0.5 (E0.5). Pregnancy was confirmed by weight gain. At the indicated time points, dams were infected s.c. with 1 × 10^4^ plaque-forming unit (PFU) of PRVABC59, euthanized at the indicated time points, and their fetuses and indicated tissues were harvested. The fetal crown rump body length (CRL), head length from the tip of the nose to the occiput, head width, and body weight were measured. For postnatal analyses, the dams were infected s.c. with 1 × 10^4^ PFU of PRVABC59 and monitored until offspring were born. The offspring were monitored every day for 14 days after birth (P14), and their body weights and head diameters were measured from P3 to P11.

For the SmithKline Beecham, Harwell, Imperial College, Royal London Hospital, phenotype assessment (SHIRPA) primary screen [50,51], dams were infected s.c. with 1 × 10^4^ PFU of PRVABC59 at E12.5, and their offspring were monitored until SHIRPA screening was performed at the indicated time points. Offspring born to PRVABC59-infected or uninfected dams were weighed at P7 or P8 to confirm their growth; they were otherwise left alone to avoid excessive handling, which may modulate the fear, anxiety, or stress response after development [52,53]. The SHIRPA primary screen was performed as previously described [51] with modifications. Briefly, each mouse was placed in a transparent cylindrical viewing jar (15 cm diameter, 11 cm height) for 5 min to observe rearing, grooming, respiration rate, and tremor. Subsequently, the mouse was transferred to an arena (33 cm wide × 55 cm long × 18 cm height) that consisted of 15 evenly spaced squares (11 cm × 11 cm) to evaluate transfer arousal and motor behavior; thereafter, palpebral closure, piloerection, gait, pelvic elevation, and tail elevation were observed in the arena. A sequence of manipulations was performed to evaluate touch escape, positional passivity, trunk curl, limb grasping and visual placing, grip strength, body tone, pinna reflex, corneal reflex, tow pinch, and wire maneuver. To complete the assessment, the mice were restrained in a supine position to record autonomic behaviors prior to measurement of the righting reflex, contact righting reflex, and negative geotaxis. Throughout the procedure, vocalization, fear, irritability, and aggression were recorded. All behaviors were scored as previously described [51]. The individual parameters assessed by SHIRPA were grouped into five functional categories: neuropsychiatric state, motor behavior, autonomic function, muscle tone and strength, and reflex and sensory function [54,55]. The neuropsychiatric state includes spontaneous activity, transfer arousal, touch escape, positional passivity, biting, fear, irritability, aggression, and vocalization. Motor behavior includes body position, tremor, locomotor activity, pelvic elevation, tail elevation, gait, trunk curl, limb grasping, wire maneuver, and negative geotaxis. Autonomic function includes respiration rate, palpebral closure, piloerection, skin color, heart rate, lacrimation, salivation, and body temperature. Muscle tone and strength include grip strength, body tone, limb tone, and abdominal tone. Reflex and sensory functions include visual placement, pinna reflex, corneal reflex, toe pinch, and righting reflex.

To induce neutralizing antibodies against ZIKV, seven female mice (Group A) and three female mice (Group B) were infected s.c. with 1 × 10^4^ PFU of PRVABC59 40 days before mating. Four female mice (Group C) or one female mouse (Group D) were infected s.c. with 1 × 10^4^ PFU of PRVABC59 twice at a 57–60 days interval before mating. Ten female mice in Group E were inoculated s.c. with 2MEM twice at a 57–60 day interval before mating. After plugging, the mice were bled and the neutralizing antibody titer was determined by the standard 50% plaque reduction neutralization (PRNT_50_) assay [56,57]. Dams in Groups A, C, and E were infected s.c. with 1 × 10^4^ PFU of PRVABC59 at E9.5, and dams in Groups B and D were inoculated s.c. with 2MEM at E9.5. The offspring were monitored from P1 to P21. For the uninfected control, seven female mice were left without any treatment before mating and during pregnancy.

### 2.7. Statistical Analyses

The Student’s *t*-test was performed for normally distributed data sets where differences in variance were <4, skewness was >−2, and kurtosis was <2. The Kolmogorov–Smirnov test was used for non-parametric data where differences in variance were >4, skewness was <−2, and kurtosis was >2. The log-rank test was used for the statistical analysis of survival rates. Repeated-measures ANOVA was used to determine differences in postnatal growth over time. Pearson or Spearman correlation analyses were performed for normally distributed data or non-parametric data, respectively. Statistical significance was set at *p* < 0.05. Statistical analysis of the experimental data was performed using JMP 13 software (SAS Institute, Inc., Cary, NC, USA).

## 3. Results and Discussion

### 3.1. Fetal Outcomes

To assess fetal outcomes after congenital ZIKV infection, dams were s.c. infected with PRVABC59 at E6.5, E7.5, E8.5, E9.5, E10.5, E11.5, E12.5, E13.5, and E14.5, and the fetuses were visually inspected at 6 days post-infection (dpi; Figure 1A). Most of the fetuses infected at E6.5, E7.5, and E8.5 (100%, 100%, and 75%, respectively) showed abnormalities (IUGR, deformed fetal/placental masses, or fetal death). The number of infected embryonic days was inversely correlated with the prevalence of IUGR or deformed masses (Figure 1B,C). The CRL (Figure 1D insert), head length from the tip of the nose to the occiput (Figure 1E insert), and weight of infected fetuses were significantly smaller than those of uninfected fetuses (Figure 1D–F). The fetal CRL and head length from the tip of the nose to the occiput were measured to provide evidence for IUGR and to predict fetal cranium growth, respectively [37,58]. The lower prevalence of fetuses with gross abnormalities after infection at or after E9.5 (Figure 1A–C) was confirmed by visual inspection at 2 and 4 dpi at E9.5 and E13.5 (Figure 1G). In addition, intracranial hemorrhage (Figure 1H) and ocular malformation (Figure 1I), which were similar to those observed in human neonates with CZS [59,60,61,62], were observed in fetuses after infection at E13.5. Thus, ZIKV infection during early pregnancy between E6.5–E8.5, corresponding to the first trimester in humans [63], was a significant risk factor for severe fetal outcomes, such as fetal death, resorption (observed as deformed fetal/placental masses), and severe IUGR, whereas infection at or after E9.5 caused relatively mild outcomes (Figure 1D–F), but did not enhance fetal lethality (Figure 1A). Thus, the embryonic timing of congenital ZIKV infection affects the severity of fetal outcomes.

### 3.2. Fetal and Placental Infection

To confirm the vertical transmission of ZIKV, viral titers in the placentas, fetal whole bodies, and deformed masses at 6 dpi were determined by CCID_50_ assays. Most placentas were infected irrespective of the infected embryonic days, whereas the titer of most fetuses was lower than the detection limit at 6 dpi (Figure 2A). To assess whether ZIKV did not transmit to fetuses or did not replicate in fetal tissues, or whether active virus replication decreased to undetectable levels before 6 dpi, the placentas and fetal tissues were collected at 2 or 4 dpi at E9.5–E10.5 (first trimester in humans) or E12.5 (second trimester in humans) and tissue virus titers determined. The fetal heads were titrated rather than the whole body, except for fetal samples collected at 2 dpi at E9.5, as the fetal heads were indistinguishable from the body. A total of 60% and 100% of fetuses in each dam infected at E9.5 and 60%, 20%, and 16.7% of fetuses in each dam infected at E12.5 were infected at 2 dpi; all fetuses were infected by 4 dpi with a similar titer after infection at E9.5–E10.5 and E12.5 (*p* = 0.97, Figure 2B). The virus titers were similar in each tissue of dams infected at E9.5–E10.5 and E12.5 (Appendix A), showing that the dams were equally susceptible to ZIKV infection irrespective of embryonic days, as previously reported [64]. The placental virus titers were not different after infection at E9.5–E10.5 and E12.5 (Figure 2B, *p* = 0.053 for 2 dpi, *p* = 0.13 for 4 dpi), with no correlation in virus titers between the fetal heads and corresponding placentas (Figure 2C, *p* = 0.47 for 2 dpi, *p* = 0.17 for 4 dpi). The placentas at 6 dpi at E8.5, E9.5, or E13.5 were smaller than uninfected placentas (Figure 2D) as previously reported [29]. However, histological abnormality was not observed in placentas from infected dams (Figure 2E,F,I,J). In addition, viral antigens were found only in the histologically normal decidual cells in the peripheral area of placentas by IHC with anti-NS1 antibody (Figure 2G,H,K,L). The mouse placenta forms a definitive structure and becomes functional around E10.5–E11.5 [65,66]. The infection of fetal heads after E12.5 infection suggests that ZIKV crossed the placental barrier. Taken together, fetuses were infected congenitally, irrespective of gross abnormalities (Figure 1) or embryonic days of congenital infection. The former observation is partially consistent with previous work in which ZIKV RNA was detected in the fetal heads with only mild IUGR after infection at E6.5 or E7.5 [29].

### 3.3. Postnatal Outcomes

To assess postnatal outcomes, dams were infected with PRVABC59 at E8.5, E9.5, E10.5, E11.5, E12.5, E13.5, and E14.5, and their offspring were monitored from P1 to P14, including measuring body weight and head circumference from P3 to P11. All offspring born to dams infected at E8.5 or E9.5, died within one day after birth (Figure 3A). The survival of offspring born to dams infected at E10.5–E14.5 was significantly lower than that of uninfected offspring (*p* = 0.0007 for E10.5, *p* < 0.0001 for E11.5, *p* = 0.0005 for E12.5, *p* = 0.042 for E13.5, and *p* < 0.0001 for E14.5) (Figure 3A). Human infants with congenital ZIKV infection are typically small for gestational age (SGA) [1] and/or exhibit failure to thrive (FTT) [16]. SGA is defined as a birth weight at least two standard deviations (SDs) below the mean for gestational age [67]. FTT is defined as subnormal growth or subnormal weight gain in infants [68]. The weight of two litters (L1 and L2) infected at E12.5, and one offspring infected at E13.5, was less than 2SD of the mean weight of uninfected offspring at P3, which was the earliest time point of weight measurement (Figure 3B). The weights of offspring infected at E10.5 or E11.5 were comparable with those of uninfected offspring (Figure 3B). There was variability in offspring weights between litters (e.g., L1/L2 versus L3 after infection at E12.5). The absence of significance in offspring weights after infection at E10.5 or E11.5 when compared with uninfected offspring thus may be explained by the small sample size, which is a limitation of our study. The mean weight gain of L1 and L2 infected at E12.5 (Figure 3C) was lower than that of uninfected offspring (*p* < 0.0001 for L1, *p* = 0.0005 for L2), suggesting SGA and FTT. Microcephaly is defined postnatally as a small head circumference ≥2 SDs of the norm [69,70]. The head circumferences of L1 and one offspring from L2 infected at E12.5, were smaller than 2SD of the mean of uninfected offspring at P3 (Figure 3D). The mean head circumferences at P3–P11 of L1 infected at E12.5, at P3, P7, P8, P9, and P10 of L2 infected at E12.5, and at P5 and P6 of L1 infected at E13.5 were smaller than 2SD of the uninfected mean (Figure 3E), suggesting microcephaly. Thus, our mouse model recapitulates multiple postnatal outcomes including stillbirths (Figure 3A), neonatal death (Figure 3A), SGA (Figure 3B), FTT (Figure 3B,C), and microcephaly (Figure 3D,E), as observed in human cases [1,13,15,17,71].

To evaluate the outcomes in grown-up mice after congenital ZIKV infection, SHIRPA primary screening was performed on 4-week-old mice born to dams infected with PRBVABC59 at E12.5. Two mice born to dams infected at E12.5, were smaller than each littermate and died at P10 before SHIRPA was performed (Figure 3F). The SHIRPA scores of infected mice were compared with two age-matched control groups: (1) 2MEM inoculation at E12.5, or (2) no treatment during pregnancy and postnatal periods. The reduced body weight of infected mice (Figure 3G) was consistent with previous data (Figure 3B,C). Two mice remained small during adulthood (10-week-old) and reached the ethical endpoint for euthanasia at 10 weeks after birth (Appendix A), although the survival between the three groups did not reach statistical significance (Appendix A). Infected offspring had significantly deficient SHIRPA scores compared with 2MEM-inoculated and/or untreated offspring in 11 tests belonging to four SHIRPA functional categories (Figure 3G): neuropsychiatric state (touch escape, positional passivity, provoked biting, irritability, and aggression), motor behavior (limb grasping and negative geotaxis), autonomic function (salivation and body temperature), and reflex and sensory functions (visual placing and toe pinch). The abnormalities in the SHIRPA screen were also confirmed in IFNAR^−/−^ offspring born to dams infected with the Natal RGN strain at E6.5. The mean body weight of one of the five litters infected with Natal RGN at E6.5 was lower than that of the uninfected litter at 3 weeks post-birth (Appendix A). The infected litter had abnormal scores in four tests: locomotor activity, tail elevation, gait, and grip strength (yellow boxes in Appendix A). Another infected litter showed an abnormality in a fifth test, namely, provoked biting (green box in Appendix A). The five tests belonged to three functional categories: neuropsychiatric state (provoked biting), motor behavior (locomotor activity, tail elevation, and gait), and muscle tone and strength (grip strength). ZIKV RNA was detected in the testis of one male offspring born to a Natal RGN-infected dam (Appendix A), confirming vertical transmission and offspring infection. Thus, our mouse model recapitulates a wide range of postnatal developmental sequelae [7,8,14,17,18].

### 3.4. Maternal Neutralizing Antibodies Prevent Offspring Outcomes

To demonstrate the utility of our model, we performed a proof of principal experiment testing whether neonatal death could serve as criteria for screening of ZIKV vaccine candidates. There are no vaccines currently available for ZIKV infection, and female mice were infected with ZIKV prior to mating to induce neutralizing antibodies, which alone were sufficient to prevent vertical transmission of ZIKV in mice [72]. The dams were infected with ZIKV at E9.5, which caused neonatal death (Figure 3A), and the survival of their offspring was monitored. The experimental scheme is shown in Figure 4A. Briefly, female mice were infected with PRVABC59 40 days before mating (Groups A and B) or infected twice with a 57–60 days interval before mating (Groups C and D). Females in Group E were inoculated with 2MEM before mating. An increased neutralizing antibody titer was detected after plugging in Groups A, B, C, and D when compared with Group E or uninfected group (Figure 4B). Pregnant mice in Groups A, C, and E were infected s.c. with PRVABC59 at E9.5, whereas pregnant mice in Groups B and D were inoculated with 2MEM at E9.5. The survival of offspring in Groups A and C was significantly improved compared with Group E (*p* < 0.0001, Figure 4C). The 1-day-old offspring in Groups A and D were visually larger than those in Group E (Appendix A). The survival of each litter infected at E9.5 (Groups A, C, and E) was correlated with the neutralizing antibody titer of each dam (*p* = 0.0011, Figure 4D). Taken together, the results demonstrated that maternal neutralizing antibodies (at least a PRNT titer of 1:10^2.8^, Figure 4B,D) prevent neonatal death in mice, and that neonatal death can serve as an in vivo phenotypic readout for screening the efficacy of candidate vaccines against ZIKV.

## 4. Conclusions

Our mouse pregnancy model recapitulated multiple fetal and postnatal outcomes seen in humans after congenital ZIKV infection. The embryonic timing of ZIKV infection affected the outcomes; infection during early pregnancy caused fetal death and severe IUGR, and infection during mid to late pregnancy caused stillbirth, neonatal death, SGA, FTT, microcephaly, or developmental sequelae. Furthermore, neonatal death in our model may be useful as a readout phenotype to evaluate the efficacy of ZIKV vaccine candidates.

## Figures and Tables

**Figure 1 viruses-13-01807-f001:**
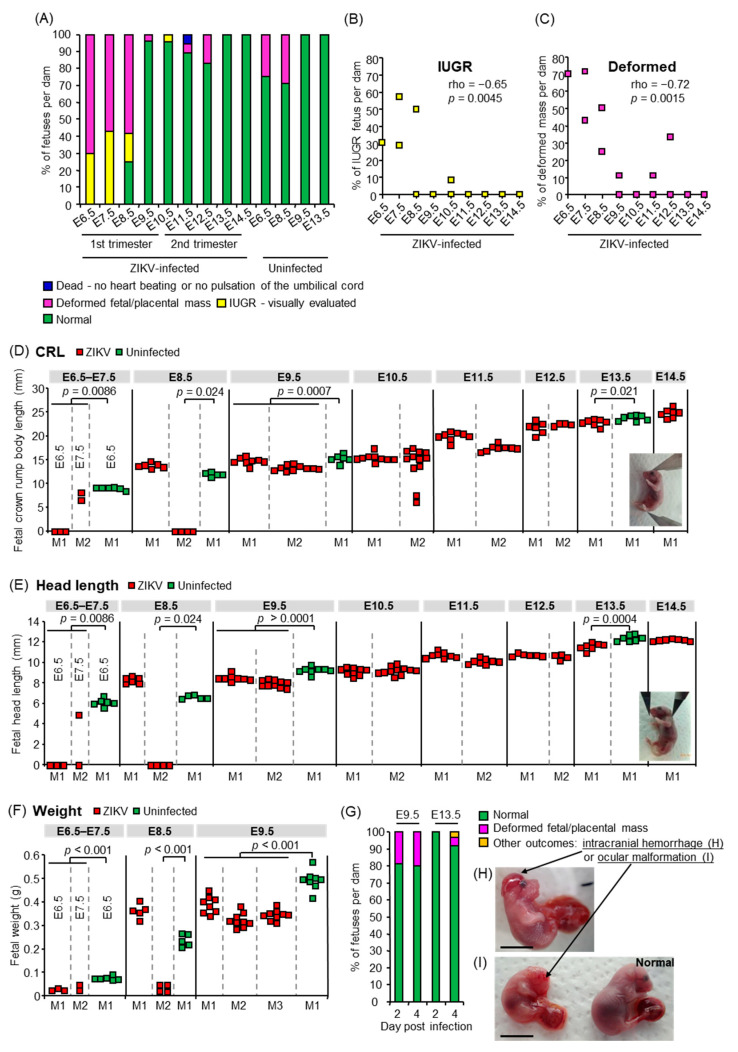
Fetal outcomes. (**A**) Percentages of each fetal outcome: fetuses that died in utero, were deformed, showed IUGR, or appeared normal at 6 days after congenital ZIKV infection. Survival of fetuses was confirmed by heartbeat or pulsation of the umbilical cord as observed under a microscope. The *x*-axis shows the embryonic days of ZIKV infection or 2MEM inoculation for uninfected controls. The ZIKV-infected group consisted of 10 fetuses from 1 dam infected at E6.5, 14 fetuses from 2 dams infected at E7.5, 22 fetuses from 3 dams infected at E8.5, 28 fetuses from 3 dams infected at E9.5, 21 fetuses from 2 dams infected at E10.5, 17 fetuses from 2 dams infected at E11.5, 13 fetuses from 2 dams infected at E12.5, and 7 fetuses from 1 dam infected at E13.5 or E14.5. The uninfected group consisted of 8 fetuses from 1 dam at E6.5 or E9.5 and 7 fetuses from 1 dam at E8.5 or E13.5. (**B**) Inverse correlation between IUGR prevalence at 6 dpi and infected embryonic days. The *x*-axis shows ZIKV-infected embryonic days. Significance was determined by Spearman’s correlation test. (**C**) Inverse correlation between the prevalence of deformed masses at 6 dpi and infected embryonic days. The *x*-axis shows ZIKV-infected embryonic days. Significance was determined by Spearman’s correlation test. (**D**) Fetal CRL at 6 dpi. Dams were infected with ZIKV or inoculated with 2MEM (uninfected) at the indicated embryonic days. Individual dams are indicated on the *x*-axis; each square represents one fetus. Vertical dashed gray lines separate litters from each dam. If the fetal heads were indistinguishable from the body, their CRL was considered as zero (below the detection limit). Significance was determined by *t*-test or Kolmogorov–Smirnov test. (**E**) Fetal head length at 6 dpi. Data are from the same fetuses as described for panel C. If the fetal heads were indistinguishable from the body, their head length was considered as zero (below the detection limit). Significance was determined by *t*-test or Kolmogorov–Smirnov test. (**F**) Fetal weights at 6 dpi. Dams were infected with ZIKV or inoculated with 2MEM (uninfected) at the indicated embryonic days. Individual dams are indicated on the *x*-axis; each square represents one fetus. Vertical dashed gray lines separate litters from each dam. Significance was determined by *t*-test. (**G**) Percentages of each fetal outcome at 2 or 4 dpi. Dams were infected with ZIKV at E9.5 or E13.5, and fetuses were visually inspected at 2 or 4 dpi. The data include 18 fetuses from 2 litters at 2 dpi at E9.5, 10 fetuses from 1 litter at 4 dpi at E9.5, 4 fetuses from 1 litter at 2 dpi at E13.5, or 75 fetuses from 8 litters at 4 dpi at E13.5. (**H**) The fetus with intracranial hemorrhage at 4 dpi at E13.5. Scale bar = 1 cm. (**I**) The fetus with ocular malformation and an apparently normal littermate. Scale bar = 1 cm.

**Figure 2 viruses-13-01807-f002:**
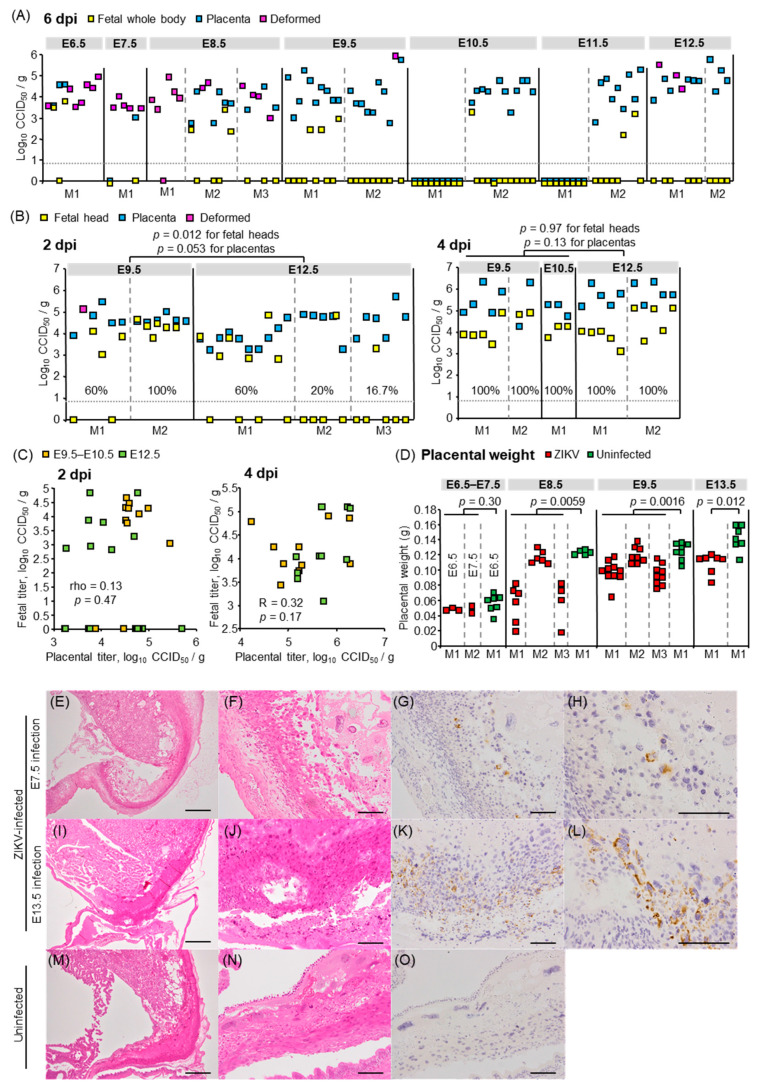
Viral titers and histological findings in fetal tissues or placentas. (**A**) Viral titers in fetal whole bodies, placentas, and deformed masses at 6 dpi. Dams were infected with ZIKV at the indicated embryonic days. Individual dams are indicated on the *x*-axis. Vertical dashed gray lines separate litters from each dam. Symbols represent individual fetus, placenta, or deformed mass. Limit of detection was 0.83 log_10_CCID_50_/g as indicated by the horizontal dashed line. (**B**) Viral titers in fetal heads, placentas, and deformed masses at 2 or 4 dpi, as described for panel A. Percentage of fetuses that were infected for each dam at 2 or 4 dpi is indicated. Kolmogorov–Smirnov test or *t*-test was used for statistical analysis. (**C**) Lack of correlation between virus titers in placentas and fetal heads at 2 dpi and 4 dpi as determined by Pearson or Spearman’s correlation test. (**D**) Placental weights at 6 dpi. Dams were infected with ZIKV or inoculated with 2MEM (uninfected) at the indicated embryonic days. Individual dams are indicated on the *x*-axis; each square represents one placenta. Significance was determined by *t*-test or Kolmogorov–Smirnov test. (**E**) H&E staining of placentas at 6 dpi; dams were infected at E7.5. Representative image of placentas from 2 dams. (**F**) As described for panel E at higher magnification. (**G**) IHC of placenta at 6 dpi at E7.5 using anti-ZIKV NS1 antibody. Positive staining (brown) was detected in decidual cells. Representative image of placentas from 2 dams. (**H**) As described for panel G at higher magnification. (**I**–**L**) H&E staining and IHC of placentas at 6 dpi; dams were infected at E13.5; otherwise as described for E–H. (**M**) H&E staining of placentas from uninfected dams. Representative image of placentas from 2 dams. (**N**) As described for panel M at higher magnification. (**O**) IHC of placenta from uninfected dams using anti-ZIKV NS1 antibody. Representative image of placentas from 2 dams. Scale bars; 500 µm (**E**,**I**,**M**), 100 μm (**F**–**H**,**J**–**L**,**N**,**O**).

**Figure 3 viruses-13-01807-f003:**
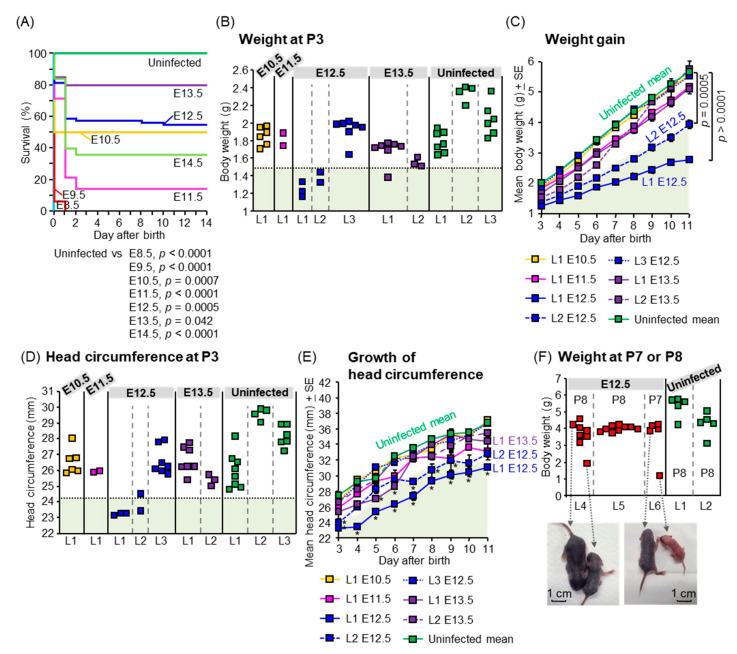
Postnatal outcomes. (**A**) Survival of offspring. Data are from 8 offspring from 1 dam infected at E8.5, 16 offspring from 3 dams infected at E9.5, 12 offspring from 2 dams infected at E10.5, 14 offspring from 2 dams infected at E11.5, 75 offspring from 9 dams infected at E12.5, 20 offspring from 3 dams infected at E13.5, 25 offspring from 4 dams infected at E14.5, and 19 offspring from 3 uninfected dams. Comparison of Kaplan–Meier survival curves between groups was performed by log-rank analysis. (**B**) Weight of offspring at P3. Individual litters are indicated on the *x*-axis; each square represents a single offspring. Vertical dashed gray lines separate each litter, which was infected at the indicated embryonic days. The pale green shaded area represents 2SD below the mean body weight of uninfected offspring. (**C**) Weight gain of each litter. The pale green shaded area represents 2SD below the mean body weight of uninfected offspring. Data consist of 6 survived offspring out of 8 offspring (6/8) for litter 1 (L1) infected at E10.5, 2/7 for L1 infected at E11.5, 3/5 for L1 infected at E12.5, 2/4 for L2 infected at E12.5, 7/7 for L3 infected at E12.5, 7/7 for L1 infected at E13.5, and 3/5 for L2 infected at E13.5. The uninfected group consisted of 18 offspring from 3 litters. Statistical analyses were performed by repeated-measure ANOVA. (**D**) Head circumference of offspring at P3. Individual litters are indicated on the *x*-axis; each square represents a single offspring. Vertical dashed gray lines separate each litter, which was infected at the indicated embryonic days. The pale green shaded area represents 2SD below the mean head circumference of uninfected offspring. Head circumference was calculated by multiplying the head diameter by Pi (3.14). (**E**) Growth of head circumference of each litter. Data are from the same litters as described for panel C. The pale green shaded area represents 2SD below the mean head circumference of uninfected offspring. Asterisks show the value 2SD below that of the uninfected mean. (**F**) Weight of offspring born to ZIKV-infected or uninfected dams at P7 or P8. Individual litters are indicated on the *x*-axis; each square represents a single offspring. Vertical dashed gray lines separate each litter, which was infected at E12.5 or uninfected. Dorsal view of two small offspring infected at E12.5 compared with each littermate. (**G**) SHIRPA scores of 4-week-old mice born to ZIKV-infected dams at E12.5. The horizontal axis shows the score. Each bar represents one mouse. Statistical analyses were performed by Kolmogorov–Smirnov test or *t*-test: * *p* < 0.05; ** *p* < 0.01. Longer lines along the *y*-axis separate litters.

**Figure 4 viruses-13-01807-f004:**
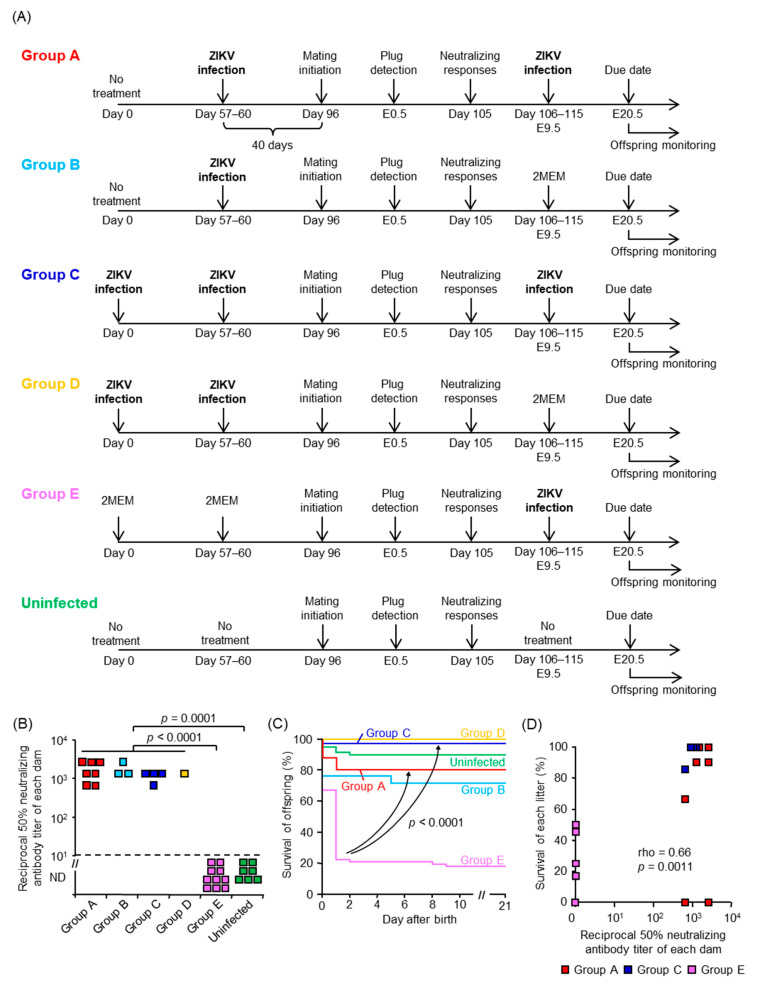
Neutralizing antibodies in dams protect the offspring from vertical ZIKV infection. (**A**) Experimental timeline of each group. (**B**) ZIKV-specific neutralizing antibody titers of each dam. Limit of detection was 1 in 10 dilutions as indicated by the horizontal dashed line. Titer was determined by PRNT_50_ assays. Kolmogorov–Smirnov test was used for statistical analysis. (**C**) Survival of offspring. Comparisons for Group E versus either Group A or Group C, *p* < 0.0001. Comparisons of Kaplan–Meier survival curves between the different groups were performed by log-rank analyses. The data are from 51 offspring from 7 litters for Group A, 21 offspring from 3 litters for Group B, 35 offspring from 4 litters for Group C, 6 offspring from 1 litter for Group D, 67 offspring from 10 litters for Group E, and 60 offspring from 7 litters for the uninfected group. (**D**) Correlation between neutralizing antibody titers of each dam and the percent survival of each litter in Groups A, C, and E. Significance was determined by Spearman’s correlation test.

## Data Availability

The data presented in this study are available on request from the corresponding author.

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
