# Peer review of "Embryonic Stage of Congenital Zika Virus Infection Determines Fetal and Postnatal Outcomes in Mice"

_viruses, 2021, doi:10.3390/v13091807_

Round 1
Reviewer 1 Report
Nakayama E. et al.
“Development of a Mouse Model Recapitulating Multiple Symptoms of Congenital Zika Syndrome”
The authors examined several timings of ZIKV infection during pregnancy. The authors found that ZIKV infection from early pregnancy period caused the fatal death, fetal resorption or severe IUGR and that ZIKV infection from mid pregnancy period led to multiple postnatal abnormalities such as stillbirths, neonatal death or microcephaly. In addition, the authors showed that adaptive immunity induced by previous ZIKV infection protected the offspring from vertical ZIKV infection. Overall, the methods and experimental designs used in this report are straight-forward and these data are novel. I have some questions/comments that the authors need to address.
- Authors concluded that neutralizing antibodies protect the offspring from vertical ZIKV infection. Although humoral immunity plays an important role in the protection, the experiments they performed could not deny the possibility that cellular immunity also has the impact on the protection. If they would like to focus on the neutralizing antibodies, passive transfer of serum from infected mice would be one of the ways to prove it. At least, the authors should comment on this point.
- Fig. 2D-N. Please add scale bars.
- Fig. 3G. Please add the statistical methods in the legend.
- Fig. 4B. Did they perform statistical analysis?
Reviewer 2 Report
In the manuscript “Development of a Mouse Model Recapitulating Multiple Symptoms of Congenital Zika Syndrome”, Nakayama et al. report a comprehensive analysis of the fetal and postnatal outcomes of ZIKV infection at different stages during pregnancy in an Ifnar1-/- mouse model. Exposure of infants to ZIKV during pregnancy results in devastating neurological and developmental abnormalities, collectively known as congenital Zika syndrome (CZS). While currently there are murine models available have for the study of the impact of ZIKV infection in embryonic development, few recapitulate the effects of CZS broadly. In this manuscript, the authors show that infection of ifnar1-/- dams with ZIKV early during pregnancy (E6.5-E8.5) has severely deleterious effects for fetal development, including deformed placental mass, IUGR, intracranial hemorrhage, ocular malformations, and low weight. In contrast, the impact of ZIKV infection on fetal development following exposure at later stages during pregnancy (E9.5-14.5) is much milder, with reduced fetal deformities and relatively normal, albeit still significantly lower compared to the uninfected fetal size and weight. The postnatal observations also reflect the increased severity of infection outcomes when exposure happens early during pregnancy. The offspring of dams infected at E8.5-E9.5 all died shortly after birth, and the survival of offspring exposed to ZIKV at E10.5-E14.5 was also lower than uninfected offspring. The SHIRPA scores in infected offspring (4 weeks old, infection E12.5) show significantly abnormal tests for neuropsychiatric state, muscle tone and strength, and motor behavior. These observations indicate that this mouse model reflects the postnatal sequela seen in infants exposed to ZIKV in utero. The study is exciting, and the experiments are well executed. However, there are concerns regarding the novelty of some of the observations in the study. Below are some recommendations for the authors:
- Title: The authors can consider adjusting the title of the manuscript to reflect the impact of their observations better. The mouse model used in the manuscript (Ifnar1-/-, C57BL/6 background) and the inoculation route (subcutaneous) were previously described by Miner et al. (Cell, May 19, 2016) and by the authors (Nakayama et al. PLoS pathogens 2021); thus it was not developed in this manuscript.
- Figure 1: In Figures 1A and 1D, only a few fetuses from one dam infected at E6.5 (1A and 1D) or E7.5 (1D) are included in the analyses. Can the authors comment on the statistical significance of the observations in these few individuals? One concern is the variability in some of the responses between the offspring of two independent dams in the same group (figure 1F, birth weight of the progeny of M1, similar to uninfected, and M2, well below uninfected).
- Line 254 “ Taken together, fetuses were infected congenitally, irrespective of gross abnormalities or embryonic days of congenital infection.” It is important to highlight that this observation has been previously described by Miner et al. in a report in which ZIKV infections at embryo day E6.5 and harvested seven days later showed the vertical transmission of ZIKV to the placenta and fetal heads.
- Fig 2D-N: Other reports of infection in ifnar1-/- dams indicate damage to the placenta following ZIKV exposure at E6.5 (Miner et al.). How do the authors reconcile these published observations with the absence of morphological changes in the placenta reported in their manuscript?
- Fig 3B: As with figure 1, the single replicates for the weight of the offspring (P3) of dams infected at E10.5 (n=6) and E11.5 (n=11) may cause some problems with the interpretation of the data. There is variability in weight between litters from dams infected simultaneously (See L1/L2 vs. L3 in E12.5 infection in this figure). Is it possible that there are effects in postnatal weight in litters from dams injected at E10.5 and E11.5 that are not seen due to the lack of replicates?
Minor points:
Please correct the typos in table S1
Round 2
Reviewer 1 Report
The authors responded to my review appropriately. I appreciate the authors' great work.
Reviewer 2 Report
Thanks to the authors for addressing all comments and questions. I do not have further suggestions.